# Louis Pasteur: Between Myth and Reality

**DOI:** 10.3390/biom12040596

**Published:** 2022-04-18

**Authors:** Jean-Marc Cavaillon, Sandra Legout

**Affiliations:** 1National Research Agency (ANR), 75012 Paris, France; 2Centre de Ressources en Information Scientifique, Institut Pasteur, 75015 Paris, France; sandra.legout@pasteur.fr

**Keywords:** antibiosis, infectious diseases, puerperal fever, spontaneous generation

## Abstract

Louis Pasteur is the most internationally known French scientist. He discovered molecular chirality, and he contributed to the understanding of the process of fermentation, helping brewers and winemakers to improve their beverages. He proposed a process, known as pasteurization, for the sterilization of wines. He established the germ theory of infectious diseases that allowed Joseph Lister to develop his antiseptic practice in surgery. He solved the problem of silkworm disease, although he had refuted the idea of Antoine Béchamp, who first considered it was a microbial infection. He created four vaccines (fowl cholera, anthrax, pig erysipelas, and rabies) in the paths of his precursors, Henri Toussaint (anthrax vaccine) and Pierre Victor Galtier (rabies vaccine). He generalized the word “vaccination” coined by Richard Dunning, Edward Jenner’s friend. Robert Koch, his most famous opponent, pointed out the great ambiguity of Pasteur’s approach to preparing his vaccines. Analysis of his laboratory notebooks has allowed historians to discern the differences between the legend built by his hagiographers and reality. In this review, we revisit his career, his undeniable achievements, and tell the truth about a hero who made every effort to build his own fame.

## 1. Introduction

In 2022, we are recognizing the 200-year anniversary of Louis Pasteur’s birth. Pasteur belongs to the pantheon of the most prestigious scientists, whose contributions allowed major improvements in the war against pathogens (Table 1). The ongoing COVID-19 pandemic reminds humanity that this war remains contemporary. However, behind the great scientist there was a man with a huge ego who made every effort to build his fame, helped by the lay press and his hagiographers (Figure 1). Among those, René Vallery-Radot [1], his son-in-law, and Émile Duclaux [2], his successor at the head of the Institut Pasteur, contributed to his legend, even if they had to tell fairy tales. Analysis of his correspondence and laboratory notebooks has allowed historians to decipher between myth and reality [3,4,5,6]. As such, a more realistic description of this character has emerged, such as that offered by Patrice Debré [7], qualifying him as unfair, arrogant, haughty, contemptuous, dogmatic, taciturn, individualist, authoritarian, careerist, flatterer, greedy, and ruthless with his opponents. This was illustrated when he was administrator and director of scientific studies at the prestigious “École Normale Supérieure” (ENS), which educates teachers and professors. His authoritarianism, his inflexible temperament, and his conflicting relationships with the students ended in the resignation of 73 students. This required the intervention of the Minister of Education and led to his resignation. Regarding his scientific contributions, Debré added: “*sometimes he gives the impression of merely checking the results described by others, then making them his own*”. One could add he was a misogynist. When Pasteur became professor and dean at the University of Lille (1854), he wrote to his rector: “*I have the honor of proposing to you that ladies no longer be admitted to the science courses of the faculty [...] I do not need to insist at length, Mr. Rector, on the inconveniences which may result from the presence of ladies at these lessons. I do not see any reason to admit them. If their number were to become large, they could cause an appreciable lowering of the level of teaching. Their presence is always a nuisance in the natural history class.*”

The acquisition of knowledge is built on the shoulders of giants; however, many of these giants owe their notoriety to more obscure scientists who opened the furrows of knowledge and sowed the seeds which would hatch in other minds. Because Pasteur’s work was disruptive with nineteenth century knowledge, he faced many opponents, and history has forgotten those scientists whose only fault was to be right ahead of him.

## 2. From Molecular Chirality to Fermentation

Born in a family of tanners, Louis was the only boy with three sisters (Table 2). He was a mediocre student who failed to pass his baccalaureate the first time, but he was an excellent pastellist. When he finally passed, he joined the ENS. Pasteur was invited by Antoine-Jérôme Balard (1802–1876), a prestigious chemist who had discovered bromine in 1826, to join his laboratory at ENS. Two other mentors supported Pasteur’s young career: Jean Baptiste Dumas (1800–1884), a professor of chemistry and member of the French Academy of Sciences, and Jean-Baptiste Biot (1774–1862), a professor of astronomy and physics and also a member of the French Academy of Sciences, who invented the polarimeter used by Pasteur for his first studies on the divergent diffraction of the light by tartaric and paratartaric acids. His studies on the optical activity and crystallography of these molecules allowed Pasteur to identify their molecular dissymmetry and their mirror-image nature. Pasteur was well ahead of his time and his discovery on molecular chirality catapulted the young Pasteur to the forefront of French research, recognizing what his eminent predecessors (J.-B. Biot, Frédéric-Hervé de la Provostaye (1812–1863), Wilhelm Gottlieb Hankel (1814–1899), and Eilhard Mitscherlich (1794–1863)) had missed [8]. For ten years, he pursued his research in chemistry and crystallography, founding stereochemistry.

While in Lille, Pasteur was contacted by beet alcohol producers who were facing difficulties in their process of fermentation. Studying this process would be the second main field of investigation of Pasteur. However, in contrast to the study of molecular chirality, Pasteur had many precursors. The fact that fermentation is part of the action of a living entity had been hypothesized since Antonie van Leeuwenhoek (1632–1723) observed yeast under his microscope in 1680. The link between these cells and the fermentation process was described in 1787 by Adamo Fabroni (1748–1816), in 1803 by Baron Louis Jacques Thénard (1777–1857), in 1836 by Theodor A.H. Schwann (1810–1882) (Figure 2), in 1837 by Friedrich T. Kützing (1807–1893), in 1838 by Pierre Jean François Turpin (1775–1840) and Charles Cagniard de Latour (1777–1859), and finally in 1854 by Antoine Béchamp (1816–1908), who understood the process a few years before Pasteur, establishing the complementarity between yeast and a soluble substance he named “zymase” (Figure 3).

With regard to the first work of Pasteur on fermentation, his text published in 1858 [9] is at the very least ambiguous concerning his position on the concept of spontaneous generation: “*It is not necessary to already have lactic yeast to prepare it: it takes birth spontaneously, with as much ease as brewer’s yeast, whenever the conditions are favorable “ […] I use this word (spontaneous) as an expression of the fact, completely reserving the question of spontaneous generation. In contact with the common air, lactic yeast is born if the natural conditions of the environment and temperature are suitable*.” In this text, he evaded the issue, hence this convoluted style. In 1856–1857, when he began to write this memoir, he had already been at work on fermentation for over a year. He was about to leave his position as dean of the faculty of Lille for the ENS. He apparently discussed this subject with his mentor Jean-Baptiste Biot, who dissuaded him from openly engaging in such a controversial subject. He knew he would need funding for his research and that he would apply for it through an Academy of Sciences Prize (Prix Montyon, 1859). He did not waver. This is what explains its lack of clarity; it does not engage yet. In addition, it was in 1860, when the Academy of Sciences proposed a competition (Alhumbert prize) to: “Try by well-made experiments to throw a new light on the question of spontaneous generations”, that he really launched into the battle and supported the role of living cells in the process of fermentation against the idea previously defended by Antoine Lavoisier (1743–1794) and arguing with his contemporary detractors, Justus Freiherr von Liebig (1803–1873), Friedrich Wöhler (1800–1882), and Claude Bernard (1813–1878). In a fight from beyond the grave, after Claude Bernard’s death, Pasteur, addressing his late opponent, published a book “*Critical examination of a posthumous writing by Claude Bernard on fermentation*” (1879), affirming *urbi et orbi* the importance of yeast and germs to obtain alcoholic fermentation.

Invited by beer and wine producers, bringing a microscope into a biochemistry laboratory, Pasteur identified pathogens that were responsible for different wine diseases. Thanks to pasteurization developed to allow the export of wines to England and his work on beer and wine, Pasteur became a recognized authority on industrial fermentation. The Whitebread breweries in Britain and Carlsberg in Denmark attribute their success to Pasteur’s visit after identifying that a microorganism was contaminating the fermentations required to make beer.

## 3. Fighting against Spontaneous Generation and Germ Theory

When he started his studies to refute spontaneous generation and initiated his germ theory, many demonstrations were already published by a large number of scientists (Table 3). Indeed, Louis Pasteur was a great admirer of Lazzaro Spallanzani (1729–1799), recognizing his immense contribution when he first demonstrated the non-existence of spontaneous generation (Figure 2). Pasteur was offered by Raphaël Bischoffsheim (1823–1906), banker, philanthropist, and deputy, a painting by Jules Édouard (1827–1878) representing Spallanzani, which hung in his large dining room. There is another scholar to whom Pasteur paid tribute by writing him in 1878: “*For twenty years now, I have been following some of the paths you have opened. As such, I claim the right and the duty to associate myself wholeheartedly with all those who will soon proclaim that you have well deserved science and to sign these few lines. One of your numerous and sympathetic disciples and admirers*” [10]. This is Theodor Schwann (1810–1882), a Berliner doctor who, in 1836, refined Spallanzani’s experiment by passing the air through a flame which enters in a flask containing an infusion sterilized by boiling. The same year, Franz Schulze (1815–1921), a German chemist and professor of anatomy in Rostock, Graz, and Berlin, enriched the experimental approach to demonstrate that air is a vector of germs. However, there were some awesome scientists who were fully ignored by Pasteur. Joseph Grancher (1843–1907), the physician who injected the first Pasteur rabies vaccine into humans, quoted some of them [11]: Jean Hameau (1779–1851) (Figure 3) was a country doctor in the southwest of France who studied glanders, malaria, dysentery, yellow fever, smallpox, and cholera. In a prophetic book, entitled “*Studies on viruses*” (1847), he explained how germs are responsible for infectious diseases. Grancher wrote: “*if M Pasteur had known his work, he would have cited him as one of his precursors*”; *Dr J. Hameau, in his study on viruses, talks about these viruses, their incubation and their multiplication, as a student of Pasteur would do nowadays. It is certainly a great accomplishment that this one! To have foreseen, divined, affirmed, with all the proofs which science of his time could offer him, a doctrine which, only fifty years later, and thanks to the genius of Pasteur, was to reign as absolute; it is, in my opinion, showing a penetrating sagacity.*” Jean Hameau died of a devastating sepsis in the arms of his son, himself a doctor. The epigraph to his book was: “*Everywhere life is in life and everywhere life devours life!*” It could not be a more appropriate definition of a microbe that carries away the human being that hosts it. Grancher also quoted Girolamo Fracastoro (1483–1553). This XVIth century doctor, who coined the word syphilis, anticipated the contagiousness of tuberculosis and considered that rabies was consecutive to the entrance of “seminaria” (germs) into the body: “*he was also an instinctive and brilliant precursor, also unknown to M. Pasteur, I am sure*.

In his fight against the concept of spontaneous generation, Pasteur was helped by Balard, who conceived the experiments with the swan neck flasks, which were decisive in demonstrating that there are germs in the air [12]. Pasteur had to fight against some strong opponents of his germ theory who continued to defend spontaneous generation. Among those were, in France, Félix Archimède Pouchet (1800–1872) and his theory of heterogenesis [13] and Hermann Pidoux (1808–1882) and his theory of organic vitalism [14] and, in the UK, Lionel Beale (1828–1906) [15]. By contrast, John Tyndall (1820–1893), a famous Irish physicist (Figure 2), was a great supporter of Pasteur’s theory of germs and published his own experiments on the presence of germs in the air [16,17]. A French catholic priest, Abbé Moigno (1804–1884), a great popularizer of science, gathered texts from Tyndall and Pasteur in a book on “*Organized microbes, their role in fermentation, putrefaction and contagion*”, [18]. The word “microbe” had been coined in 1878 by a French surgeon, Charles Sédillot (1804–1883), as a tribute of the works of Pasteur: “*Mr. Pasteur has demonstrated that microscopic organisms, widespread in the atmosphere, are the cause of the fermentations attributed to the air, which is only the vehicle and has none of their properties […] The names of these organizations are very numerous and will have to be described and, in part, reformed. The word microbe having the advantage of being shorter and of a more general meaning, and my illustrious friend M. Littré, the most competent linguist in France, having approved it, we adopt it […]*” [19]. However, Pasteur preferred to use the word microorganism. The word bacterium was coined in 1828 by Christian Gottfried Ehrenberg (1795–1876), a German naturalist, from the Greek βακτηριον, meaning “little stick”. The same six species of *Vibrio* described by Ehrenberg were recognized 35 years later by Pasteur as being the germs of putrefaction [20].

## 4. Fighting against the Silkworm Disease

Alexander von Humboldt (1769–1859), the great explorer, stated: “*The frequent sequence of reactions to an important discovery is first a denial of its veracity, then a denigration of its importance, and finally usurpation of credit for it*”. Such a statement fully fits with the contribution of Pasteur in the fight against silkworm disease. J.B. Dumas was minister of agriculture and trade and a senator in 1865 when he invited his former student to address the major threat that was facing the silkworm industry. Pasteur received subsidies from the government and spent five stays in Alès in the Cévennes. Pasteur initially admitted that he knew nothing about this topic. The famous entomologist Henri Fabre said: “*Ignoring caterpillar, cocoon, chrysalis, metamorphosis, Pasteur came to regenerate the silkworm. The ancient gymnasts presented themselves naked to the fight. Genius fighter against the scourge of the magnaneries, he also came to the battle naked, that is to say, without the simplest notions about the insect to get out of the peril. I was stunned; better than that I was amazed*” [21].

For two years, Pasteur denied that the disease (pebrin) could be due to a pathogen. However, Agostino Bassi (1773–1856), an Italian entomologist, had demonstrated as early as 1835 that another disease (muscardin) was caused by a fungus (*Beauveria bassiana*). In 1844, Bassi asserted the idea that not only animal (insect) but also human diseases are caused by other living microorganisms. On his side, A. Béchamp (Figure 3), without official support, suggested as early as 6 June 1865 in front of the Central Agricultural Society of Hérault that the disease was due to a parasitic pathogen. On 25 September 1865, Pasteur communicated to the Academy of Sciences, opting for a spontaneous intrinsic blood disease [22]. The following year, Béchamp published: “*The pebrin, in my opinion, attacks the worm from the outside first, and the germs of the parasite come from the air. Sickness, in short, is not originally constitutional*” [23,24]. Béchamp’s diagnosis was supported by Édouard-Gérard Balbiani (1823–1899), an entomologist and embryologist who declared in 1866: “*The corpuscles that we observe in the disease described under the name of pebrin in silkworms are not anatomical elements […] but indeed psorospermia, that is to say parasitic plant species*” [25]. It was Désiré Gernez (1834–1910), working alongside Pasteur, and Franz von Leydig (1821–1908), a German zoologist, who had been contacted by Pasteur, who both convinced Pasteur of the real nature of disease. Gernez had worked as a physics associate/preparer in Pasteur’s laboratory at ENS from 1860 to 1864, and he joined Pasteur in Alès. The same as Béchamp and Balbiani, he concluded that the disease was parasitic. It was only in April–May 1867 that Pasteur sent a letter to Dumas finally acknowledging the parasitic origin of the disease [26]. Pasteur sent a letter in 1867 to the secretary of the agricultural committee, which overwhelmed Béchamp with discourteous contempt: “*Poor Mr. Béchamp is at this moment one of the most curious examples of the influence of preconceived ideas gradually turning into fixed ideas. All his statements are so biased that I wonder if he has ever observed more than ten silkworms in his life*” [27]. In his book written in 1870 on his studies of the diseases of silkworms, dedicated to the Empress Eugénie, to better capture all the glory and build his legend, Pasteur neither admitted his wanderings nor acknowledged the visionary works of Béchamp. The latter said: “*I am Pasteur’s forerunner, just as the stolen is the forerunner of the fortune of the happy and insolent thief who taunts and slanders him*” [28]. Unfortunately for Béchamp, his approach to treat the disease with fumigations of creosote was not fully appropriate while Pasteur’s technique of segregating the cocoons, an approach already proposed by Emilio Cornalia (1824–1882), an Italian naturalist, was successful. Thus, it was Pasteur who put an end to the epidemic and reaped all the praise. In fact, the success was very partial, as the production of cocoons, which reached 25,000 tons per year in 1850 and had collapsed to 5000 tons in 1865, never exceeded more than 8000 tons by the end of the 19th century.

Sadly for Béchamp, his concept of “microzyma”, which he subsequently developed, did not contribute to letting him enter the pantheon of heroes of microbiology. According to him, any animal or plant cell would be made up of small particles capable, under certain conditions, of evolving to form “microzymas”, small autonomous elements which would continue to live after the death of the cell from which they would come. After Pasteur’s death, Béchamp published a booklet entitled: “Louis Pasteur—His chemicophysiological and medical plagiarism—His statues” (1903). In this work, Béchamp published his various letters written in vain to restore the truth to the directors of the “*Petit Journal*” and “*La Liberté*”. He denounced “*The most brazen plagiarist of the nineteenth century and of all centuries: it is Pasteur*” and criticized the press “*for propagating the false legend which makes a famous plagiarist a great man*”. Reading Béchamp, one sympathizes with so much suffering illustrated by such harsh words: “*Pasteur, great man, the purest glory of the nineteenth century and undisputed scholar, not only he was not, but the pure truth is that he was the less genius, the most simplistic and the most superficial scientist of our time, at the same time the most plagiarist, the most false, and the biggest noise-maker of the nineteenth century*” and having shamelessly attributed the success to himself, Pasteur was able to further promote himself to Dr. Paul Bert (1833–1886), student of Claude Bernard and member of the National Assembly, where he obtained for Pasteur by a vote on 28 March 1874 a life pension of 12,000 gold francs per year, transformed on 13 July 1883 into a pension of 25,000 francs.

## 5. Identifying the Germs of the Infectious Diseases (1877–1881)

One of the main handicaps of Pasteur was having been educated as a physicist and a chemist and thus having ignored some key contributors in the field of medicine, infectious diseases, and physiology of inflammation. Furthermore, because he only spoke the French language, he missed many breakthrough publications from German scientists, leading him to make incorrect statements. For example, in 1878, he claimed [29]: “*For us currently, it would be the red blood cells that would be the pus cells from a simple transformation from the first into the second*”, ignoring the work of two of Rudolf Virchow’s (1821–1902) students, Julius Friedrich Cohnheim (1839–1884), who, eleven years earlier, had demonstrated that white blood cells cross blood vessels to become pus cells [30], and that of Julius Arnold (1835–1915), who in 1875 had illustrated the diapedesis of blood cells [31].

The competition between the German school led by Robert Koch (1843–1919) (Figure 2) and that of Pasteur [32] was based on the identification of the germs responsible for some infectious diseases. Of course, the name of Koch is associated with discovery of the bacillus of tuberculosis and improperly to that of cholera, which was first identified by Filippo Pacini (1812–1883) in Florence (Italy) in 1854. The name “*Pasteurella*” was inappropriately coined by an Italian bacteriologist in 1887, Vittore Trevisan (1818–1897), while the germ responsible for the fowl cholera was first identified by two Italian scientists, Sebastiano Rivolta (1832–1893) in 1877 and Edoardo Perroncito (1847–1936) in 1878! In France, the first isolation of this bacterium was made by Henri Toussaint (1847–1890) in 1879 (Figure 3), a medical doctor, a veterinarian, and a docteur ès-Sciences who provided Pasteur with this germ.

Let us speak of Joseph Grancher [11], the first French medical doctor with Isidore Strauss (1845–1896) who, thanks to Émile Roux (1853–1933), studied bacteriology with Pasteur under the supervision of Charles Chamberland (1851–1908): “*But already Germany had surpassed us in microbial techniques and Mr. Pasteur’s laboratory, faithful to cultures in liquid media, neglected the art of staining microbes and that of cultivating them on solid media. It was M.*
*Babès (Victor Babeș (1854 in Vienna–1926 in Bucharest)) who, coming from Germany, introduced in France, in M. Cornil’s laboratory (Victor André Cornil (1837–1908)), the methods of staining microbes then used in M. Koch’s laboratory. And I believe I brought from Berlin, after a trip made with M. Brouardel (Paul Brouardel (1837–1906)) for Emersleben trichinosis (in November 1883), the first tubes of gelatinized blood serum.*” Indeed, thanks to the work of Angelina (Fanny) Hesse (1850–1934), her husband Walther Hesse (1846–1911), and Julius Richard Petri (1852–1921), Koch’s team had developed the agar–agar containing culture medium and the device known as the Petri dish. Grancher, with great objectivity rare in the close entourage of Pasteur, recognized the superiority of the experimental approach of the German school of bacteriology. The consequences were expressed in terms of discoveries: “*And while the studies on rabies and the search for its microbe continued on rue d’Ulm, while a few French doctors were beginning or better resuming their studies, Germany gave us almost in quick succession the important discoveries of the microbe of erysipelas, diphtheria, glanders, tetanus, pneumonia, this one recognized at the same time in France by Talamon (Charles Talamon (1850–1929)*” [33].

However, studying the boils of his colleague É. Duclaux and samples from a 12-year-old girl suffering from osteomyelitis, Pasteur identified *Staphylococcus* in 1880 [34] concomitantly with Alexander Ogston (1844–1929), a British surgeon who was studying the germs present in abscesses [35]. Ogston indicated that 917,775 cells/mm^3^ were present in pus, which contained 2,121,070 micrococci/mm^3^. In 1882, Ogston coined the word *Staphylococcus* from ancient Greek staphyle, which means a bunch of grapes. Furthermore, concomitantly with the American George M. Steinberg (1838–1915) in 1881, Pasteur identified the bacteria first known as *pneumococcus*, then *diplococcus pneumonia*, and finally named *Streptococcus pneumoniae* [36,37].

### 5.1. Puerperal Fever

Pasteur investigated puerperal fever ten years after Victor Feltz (1835–1893) and Léon Coze (1819–1896), two physicians working in Strasbourg, demonstrated in 1869 the presence of a deadly bacterium (*Streptococcus*) in the blood of a patient who died of puerperal fever [38]. Starting in 1865 and for four years, the two Alsatian doctors established the presence of contaminating germs able to transmit death to rabbits injected with the blood of patients with typhoid fever, smallpox, pneumonia, erysipelas, and scarlet fever [39]. On 17 March 1879, Feltz, then in Nancy after the loss of Alsace in the 1870 war, republished a similar observation in the Comptes Rendus de l’Académie des Sciences, although this time he transmitted the death to a guinea pig [40]. Feltz called his observed bacteria *Leptothrix puerperalis.* The following day, Pasteur reported in the Bulletin de l’Académie de Médecine the presence of germs in the lochia, blood, and uterus from a patient who died of puerperal fever [41]. Pasteur did not perform experiments to transmit the disease to an animal but suggested washing the genital tract with diluted boric acid. Pasteur came into contact with Feltz and denied Feltz’ observation. He obtained some blood of Feltz’ patient, which he injected into one guinea pig while two others were injected with anthrax. He sent the animals by train to Nancy, where Feltz received the dying guinea pigs. Then, amazingly, Feltz, the medical doctor, accepted the diagnosis given by the scientist and he conceded that his patient after delivery had died of anthrax despite there being no case of anthrax in the area [42]! Most probably, both had observed *Streptococci*, a name coined by Theodor Billroth (1829–1894), as a combination of the ancient Greek streptos meaning twisted and kokkos meaning berry.

In the movie “*The story of Louis Pasteur*” (1936), directed by William Dieterle with Paul Muni playing Pasteur (a role for which he received the Oscar for best actor), following one death after puerperal fever, a document was created stating “*Wash your hands. Boil your instruments. Microbes cause disease and death to your patients*”, signed Louis Pasteur. In fact, Pasteur never mentioned that the hands of the obstetricians could transmit the disease, although Pasteur was very reluctant to shake hands and was himself regularly washing his hands. However, Pasteur and the scriptwriter ignored the statements of Alexander Gordon (1752–1799), who admitted in 1795 that he had transmitted diseases to women after delivery [43], and the major achievement of Ignaz Semmelweis (1818–1865), who demonstrated in 1847 that the hands of medical students, after performing autopsies, contaminated the parturients they visited [44]. In his Vienna hospital, Semmelweis advocated hand and nail washing with calcium hypochlorite, reducing the mortality from 16% to 0.85% [45].

### 5.2. Anthrax

The *Bacillus anthracis* was first observed in Germany by Aloys Pollender (1799–1879) in 1855 and Friedrich A. Brauell (1807–1882) in 1857. In France, in 1850, Pierre Rayer (1793–1867) was the first to demonstrate the contagiousness of the disease. However, the main achievement was accomplished by a precursor of Pasteur, Casimir J. Davaine (1812–1882) (Figure 2). Jean Rostand (1894–1977), a famous writer and biologist, wrote: “*It is commonly believed in the public that it was Pasteur who discovered the role of microbes in the production of infectious diseases. In fact, this considerable discovery does not belong to him; it belongs to another French scientist: Davaine […] Who, the first, dared to affirm and knew how to demonstrate by the experimental method that a microscopic organism is the agent responsible for a disease*” [46]. Similarly, Jean Theodoridès (1926–1999), one of the most prestigious French historians of biological science wrote: “*The credit of demonstrating for the first time the pathogenic role of a bacterium in the human being and in domestic animals goes to the little-known French physician Casimir Davaine*” [47]. In 1863, Davaine observed the presence of bacteria in the blood of animals with anthrax and showed that the disease was communicable by infected blood [48]. Later, he reported that only live bacteria can transmit the disease [49]. Davaine was the first scientist to make a direct link between the presence of certain bacteria and an infection. He was well aware of the importance of his contribution: “*It has been a long time since doctors or naturalists theoretically admitted that contagious diseases, serious epidemic fevers, plague, etc., are determined by invisible animalcules or by ferments, but I am not aware of any clear demonstration to confirm this view*”. Indeed, Davaine was recognized by Pasteur to have been a major predecessor of his own work. In 1876, Koch was the first to publish photos of anthrax [50].

In 1877, Pasteur and Jules Joubert (1834–1910) reported [51] that the bacterium of anthrax could not develop when associated with other microorganisms: “*life prevents life*”. This was the very first report of a phenomenon named “antibiosis” by Jean Paul Vuillemin (1861–1932), mycologist and professor at the faculty of medicine in Nancy in 1889. The phenomenon would give rise years later to the discovery of the antibiotics. Finally, in 1880, Pasteur offered up an explanation for the natural contamination of cattle. He demonstrated that earthworms brought germs emanating from the carcasses of the dead sick animals to the surface, which had been buried in fields [52].

### 5.3. Cholera

In August 1883, a cholera epidemic broke out in Alexandria (Egypt). On August 15th, Pasteur sent his collaborators, Émile Roux, Isidore Strauss, Edmond Nocard (1850–1903), and Louis Thuillier (1856–1883), to isolate the germs and to reproduce the disease in animals. Not only did the mission fail, but Thuillier contracted cholera and died. Koch, on 24 August, also traveled to Alexandria to isolate the bacillus from the intestinal mucosa of dead people. He pursued his travel to Calcutta, India where another epidemic had broken out. On 7 January 1884, he sent a telegram informing Berlin that he finally isolated and cultured the bacillus. In 1893, Pasteur entrusted André Chantemesse (1851–1919) with a mission in Constantinople for another cholera epidemic. There, Chantemesse organized the fight against the epidemic with the construction of three disinfection stations. Later, Institut Pasteur sent Waldemar Haffkine (1860–1930), who trained in Metchnikoff’s laboratory, to fight against cholera epidemics in India thanks to a vaccine he had developed [53].

### 5.4. Plague

At the request of Institut Pasteur, Alexandre Yersin was sent to Hong Kong in 1894 to study the nature of the plague epidemic that was raging there. He was in competition with Shibasaburō Kitasato (1853–1931), a former trainee of Koch. When Kitasato was looking in blood samples, Yersin was luckier in studying buboes. On 20 June 1894, Yersin isolated the bacillus responsible for the disease, later named *Yersinia pestis* [54]. Back in France, he developed with Emile Roux, André Borrel, and Albert Calmette an anti-plague horse serum. While a large plague epidemic occurred in Guǎngzhōu (China) in 1896, Yersin went there to successfully offer his anti-plague serum. In the following years, Haffkine developed an anti-plague vaccine used in India to fight plague epidemics [53].

## 6. Pasteurization, Filtration and Sterilization

As previously mentioned, in his fight against the diseases of wines, Pasteur obtained a patent on 11 April 1865 that offered a means to get rid of the contaminating bacteria by heating the wines at 64 °C for 30 min. This process was later adapted to other products and named pasteurization. However, this approach had been previously proposed by Alfred de Vergnette de Lamotte (1806–1886), a gentleman winemaker (1846) of whom Pasteur denied the anteriority of his work. However, in fact, the very first approach had been reported in 1831 by Nicolas Appert (1749–1841), inventor of preserves who proposed the heating of wine in the 4th edition of his book [55] Then, Pasteur offered a scientific explanation to the empirical findings of his predecessors. Of note, it was Franz von Soxhlet (1848–1926) who first applied the process to milk.

In Pasteur’s laboratory, his close collaborator Charles Chamberland defended his doctoral thesis in 1879 on the origin and development of microorganisms. This was the starting point for his work on the sterilization of culture media that led him to design a disinfection oven that bears his name: the Chamberland autoclave. In 1884, to fight against the spread of typhoid fever raging in Paris, he developed a filter, designed from a porous porcelain of his invention, to eliminate microbes from drinking water. The instrument was named the Chamberland filter–Pasteur system and became very popular to provide safe drinking water. A Pasteur–Chamberland filter company was created in Dayton, Ohio, USA. They sold germ-proof filters to private homes, hotels, bars, and restaurants, offering many different designs. They advertised: “*This filter was invented in my laboratory, where its great usefulness is put to test every day. Knowing its full scientific and hygienic value, I wish it bears my name. Louis Pasteur*” [56].

Pasteur’s discoveries on germs allowed great advances in the practice of surgery. In 1865, Joseph Lister (1827–1912), a Scottish surgeon in Glasgow (Figure 2), learned Louis Pasteur’s theory that microorganisms cause infection. Using phenol as an antiseptic, he reduced the mortality of amputee patients to 15% in four years, compared to 45–50% who died of sepsis previously. He is considered to be the founder of antiseptic medicine [57,58]. In 1870, Alphonse Guérin (1816–1895), a French surgeon, invented the wadded bandage and declared: “*I firmly believed that miasmas emanating from the pus of the wounded were the real cause of that dreadful disease to which I had had the pain of seeing the wounded succumb [...]. I then had the thought that the miasmas whose existence I had admitted because I could not otherwise explain the production of the purulent infection, and which were known to me only by their deleterious influence, might well be animated corpuscles that Pasteur had seen in the air [...] If the miasmas were ferments, I could protect the wounded against their fatal influence by filtering the air as Pasteur had done [...] I imagined then the wadded bandage and I had the satisfaction to see my forecasts being carried out*” [59].

On 27 December 1892, for Pasteur’s 70th birthday, the international scientific community celebrated Pasteur’s “jubilee”. The reception took place in the large amphitheater of the Sorbonne. In a painting painted ten years later, the artist Jean-André Rixens recalled this celebration displaying Lister precisely in the middle of the painting, shown going up a few steps to congratulate Pasteur (Figure 4). In 1874, Just Lucas-Champonnière (1843–1913), after travelling to Scotland, introduced Lister’s antiseptic approach in France [60]. Similarly, Lewis Atterbury Stimson (1844–1917) attended a presentation of Pasteur in 1875 at the Academy of Medicine on spontaneous generation and the capacity of lime hyposulphite to instantly destroys all germs. Back in New York, in January 1876, he successfully completed the first amputation in the USA under completely aseptic conditions [61].

## 7. Elaboration of Four Vaccines

### 7.1. Fowl Cholera

Toussaint sent the heart of a guinea pig inoculated with the germ of chicken cholera to Pasteur in December 1878. After Pasteur had obtained the *Pasteurella* from Toussaint, he prepared a bacterial culture and developed his first vaccine against fowl cholera, which he reported in 1880 [62]. The legend told by Duclaux [2] is the following: a virulent culture of *Pasteurella* that was killing injected hens was left on the bench during Pasteur’s vacation. Back from vacation, Pasteur used this bacterial culture, which failed to kill the hens. He prepared a newly fresh virulent culture, injected it in the same hens, and these hens survived the lethal injection. From that observation, Pasteur elaborated that bacteria exposed to air or oxygen lose their virulence and can be used as a vaccine. Then, it could be claimed: “*In the fields of observation, chance favors only prepared minds*”. However, this event never happened. In 1878, Pasteur asked his son-in-law to never show his laboratory notebooks to anyone. However, in 1964, his grandson, Professor Louis Pasteur Vallery-Radot (1886–1970), donated the 152 notebooks to the French National Library, allowing the historians to explore legend and reality. The most accomplished investigation was carried out by Gerald L. Geison (1943–2001), who was awarded [3] with the William H. Welch Medal by the American Association for the History of Medicine for his book. The demystification of the great hero by Geison led to numerous laudatory comments [63,64]. The book was judged to be judicious, meticulous, and carefully argued [65]. Only the chapter on molecular chirality was severely criticized [66]. Jean Théodoridès wrote: “*This critical but objective work demystifies Pasteur, who became, partly on his own initiative, a hero of his time and a concentrate of all human virtues*” [67]. In addition, he recalled Auguste Lutaud (1847–1925), one of the most virulent Pasteur opponents: “*In France, one can be anarchist, communist, or nihilist, but not anti-Pasteurian. A simple scientific question has become a matter of patriotism*”. Théodoridès regretted that Geison did not address the story of silkworms nor the “Rouyer affair”.

Similar critical analyses have been previously proposed by Antonio Cadeddu [4,5,6] and Philippe Decourt (1902–1990) [27]. Considering the publications of Pasteur, his correspondence, the book of Pasteur’s nephew, Adrien Loir (1862–1941) [68], and the laboratory notebooks, they showed how much Pasteur sought for glory above all, to the detriment of his predecessors and his collaborators, rigging his experiments if necessary or distorting his results.

The analysis of notebook #88 reveals no text between July 1879 and November 1879: Pasteur was on vacation in Arbois, where he celebrated the wedding of his daughter Marie-Louise to René Valéry-Radot and afterward suffered from a gastroenteric disease. Texts from mid-November deal with anthrax, boils, and puerperal fever but not with the fowl cholera vaccine. On 14 January 1880, Pasteur wrote in his lab book: “*Hen’s germs: when should we take the microbe, so it could vaccinate?”,* illustrating that there was not yet a clear understanding of the protective vaccine.

### 7.2. Anthrax (1881)

While the story of the fowl cholera vaccine was romanticized, Pasteur and his team were among the first after Edward Jenner to propose a new vaccine. However, the glory of developing the first vaccine against anthrax should not be attributed to Pasteur. In August 1880, Toussaint published his efforts to attenuate germs to obtain a protective vaccine in dogs and sheep [69]. He tried heating the bacteria, which was not successful; however, treating the germs with phenol led to a protective vaccine. In Vincennes, in August 1880, Toussaint organized a vaccination session on a total of 26 sheep on the farm of the Alfort Veterinary School in Vincennes. Twenty-two animals successfully resisted to the anthrax challenge [70].

The Pasteur team was groping for the development of an anti-anthrax vaccine. The main problem was Pasteur’s belief that it was exposure to air that produced attenuated germs that should be used for vaccinations. Chamberland and Roux tried various approaches with heated blood exposed or not to oxygen or attenuated by an antiseptic. About this last approach, Pasteur said to them: “*Me alive, you will not publish this, until you find the attenuation of the bacterium by oxygen. Look for it!*” [68]. However, Pasteur surprised his two acolytes when he announced in April 1881 that he had accepted the proposal of Charles-Paul-Marie Moreau, baron de La Rochette (1820–1889), president of the Society of Agriculture of Melun: “*We put at your disposal 60 sheeps. Ten will not undergo any treatment, 25 will be vaccinated, 25 will not be. After 12 new days, we will inoculate the virulent strain of the disease to the 25 sheeps and 25 others who did not receive a vaccine. Then we will see the results*.” Taken aback, Chamberland and Roux were preoccupied and very busy actively pursuing the tests. The experiment was carried out on the farm of the veterinarian Joseph Hippolyte Rossignol (1837–1919) in Pouilly-Le-Fort in the presence of many personalities, including Eugène Tisserand (1816–1888), veterinarian and director at the Ministry of Agriculture, and a few journalists including one from *The Times*, who came from London specially to attend this unprecedented event. The vaccine was administered on 5 May 1881, as announced according to the protocol; however, two goats replaced two sheep, and eight cows, an ox, and a bull were added to the experiment, although the Pasteur team did not have any expertise with cattle. A boost was performed twelve days later. On 31 May, the very virulent strain was injected into all vaccinated and control animals. On June 2nd, all these people were back in Pouilly-Le-Fort. It was a huge success; the vaccinated sheep were in great shape, except for one ewe that died. It was identified that she was pregnant and had a stillborn fetus in her womb. The control animals were all dead or dying when the public rushed to the experiment site. All vaccinated and naïve cattle survived the inoculation. Baron de la Rochette and Dr. Rossignol hailed Pasteur’s great victory over anthrax. The phrase “*Fortune favors the daring*” has never been applied so well. Pasteur made an incredible bet, while his vaccine was still in its infancy, that no previous experiment had been conducted on this scale, and he descended into the arena, inviting the public to witness his experiments live. On 13 June 1881, Pasteur communicated his brilliant results to the Academy, failing to specify the nature of his vaccine [71], and for a good reason. In their race to produce an effective vaccine on time, Pasteur, Roux, and Chamberland ended up adopting Toussaint’s approach, namely, to attenuate the virulent germ, not by exposure to air as Pasteur will continue to imply but by exposing it to an antiseptic agent, in this case potassium dichromate.

Of course, the Pouilly-Le-Fort event caused a great sensation, and a statue was commissioned from the sculptor André d’Houdain and erected in 1897 in Melun. The bronze statue would eventually be melted down in 1943, the Vichy regime offering the occupier something to make cannons. However, a controversy arose between the professors of the Turin Veterinary School and their director, Domenico Vallada (1822–1888), to whom Pasteur had sent his vaccine against anthrax. Unfortunately, this vaccine was unable to protect Italian sheep. Pasteur estimated that the Italian veterinarians had made the mistake of inoculating for the test the blood of a corpse that had been dead of anthrax for more than twenty-four hours and that consequently germs other than those specific to anthrax had been injected. Vallada and his colleagues in Turin responded by publishing a text entitled “*On the scientific dogmatism of the illustrious Prof. Pasteur and the use that can be made of it*” (10 June 1883) [72]. They concluded their text to charge: “*We do not want to take away from our illustrious opponent the illusion of complete success which may have smiled upon him in this discussion, we likewise refrain from disturbing the sweet pleasure he experienced, when he provided new proof of the fault committed by the Turin Commission, however we believe that we are not straying from the truth, and not disrespecting him either, by expressing the opinion, that his complete success may in some way be compared to the historic victory of Pyrrhus*”. This was not the only failure of the vaccine. Nikolaï Gamaleïa (1859–1949), a medical doctor from Odessa who had come to Paris to be trained by the Pasteurians, studied the parameters that influenced the preparation of the anthrax vaccine and reported his own experiments carried out on more than 300 sheep and some dogs, rabbits, and rats [73]. He reported that certain preparations could kill sheep and established that the fever induced by the vaccine was a prerequisite for its effectiveness. Despite his efforts to master a vaccine that was complicated to prepare, during the summer of 1887, an anthrax vaccination organized by the Odessa bacteriological station resulted in the death of 80% of the vaccinated animals, i.e., 3549 sheep, at a cost of more of 40,000 rubles. The owner asked Elie Metchnikoff (then director of the bacteriological station) and Gamaleïa to reimburse him half the price and started a lawsuit. Of course, the popular press echoed this disaster. No doubt a major mistake had been made, in particular the large-scale use of a vaccine not previously tested. By contrast, Adrien Loir organized a successful anthrax vaccination of 400,000 sheep in Australia.

Pasteur reported his discoveries on the vaccine against fowl cholera and anthrax at the International Medical Congress in London (1881), where he stated: “*I have given the term vaccination a broad meaning. I hope science will dedicate it as a tribute to the merit and immense service rendered by one of England’s greatest men, your Jenner. What a joy for me to glorify that immortal name on the very soil of the noble and hospitable city of London*”. In fact, the word vaccination was coined in 1800 by Dr. Richard Dunning (1761–1851) [74], a founding member of the Plymouth Medical Society and friend and great supporter of Jenner, who had endorsed the word. As testimony of his admiration, Dunning named one of his sons Edward Jenner Dunning. Unfortunately, the child died at the age of ten months. Pasteur’s talk on the attenuation of viruses at the Fourth International Congress on Hygiene and Demography in Geneva (1882) ended with a vehement reply from Koch [75]: “*Pasteur is not a physician, and he cannot be expected to be able to comment accurately on pathological processes and symptoms of the disease.” […] ” The tactic followed by Mr. Pasteur is to communicate only what speaks in his favor about an experiment, and to ignore the facts which are unfavorable to him even when those are decisive for the purpose of the experiment. Such methods may be appropriate when it comes to advertising in business, but science must vigorously reject them. “ […] “It i**s not only by the flawed methods, but also by the means of publishing his research, that Pasteur has provoked criticism. In industrial enterprises, it is permissible, and often even in commercial interests, to keep the process that led to the discovery a secret. But in science, it is another habit which is applied. Whoever appeals to faith and confidence of the scientific world has the duty to publish the methods that it follows, in such a way that everyone is able to verify the accuracy of the published results. M. Pasteur does not comply with this duty. Already in his publications on chicken cholera, Mr. Pasteur has long hidden his method of attenuating the virus and finally it was only at Colin’s (Gabriel-Constant Colin (1825–1896), Professor at the Maison Alfort veterinary school, Member of the Academy of Medicine) insistence that he decided to publicize his method. The same was repeated about the mitigation of the anthrax virus, because the communications that Mr. Pasteur has made so far on the preparation of the two vaccines are so imperfect that it is impossible without further information to repeat and examine its process*”. About Colin, Pasteur said: “*Only one path leads to the truth, a thousand lead to error, but it is always one of the latter that Mr. Colin chooses*” [76]. On his turn, Pasteur argued against Koch.

### 7.3. Pig Erysipelas

The development of Pasteur’s third vaccine was undoubtedly the least controversial. The originality of this work, however, is that Pasteur discreetly abandoned his approach attenuating germs by exposure to oxygen in the air [77]. In the summer of 1877, Achille Maucuer (1845–1923), a veterinarian based in Bollène (Vaucluse), wrote to Pasteur to challenge him on a pathology that was rampant in pig farms, swine erysipelas. Pasteur admitted he had never heard about that disease and asked Maucuer to provide him with some documents. In his letter (23 September 1877), Pasteur wrote a diatribe on the organization of research in his country that has a very particular resonance, as in recent years (2010–2020) France has dropped from fifth to ninth place in terms of scientific production in biology and medical sciences: “*If I could master my material resources for the research projects which impassion me, I would train young scientists who, under my direction, would undertake studies on all the contagious diseases of animals and men; but our poor France, always grappling with politics, remains ignorant of the great destinies of science. I would like to see the public authorities ceaselessly preoccupied with scientific interests; most often it is for immediate utility that they consider them. Witness, in the subject which occupies us, the standing committee of epizootics decreed in 1876 and which until now limited its work to the laws of sanitary police; without trying anything for the knowledge of the epizootics*” [78].

Pasteur wished to have access to the bacillus responsible for the disease. Maucuer sent a sick pig who died at Lyon Perrrache railway station. Nevertheless, it allowed the first inoculations. The germ, *Erysipelothrix*, was first isolated by Robert Koch in 1876–78 from septic mice that had been inoculated subcutaneously with the blood of rotten meat. In 1882, Friedrich Loeffler (1852–1915) observed a similar organism in the skin blood vessels of a pig that died of porcine erysipelas and published the description of the organism a few years later. Pasteur entrusted his assistant Louis Thuillier with the task of isolating the germ. The success of the young assistant was made possible by the development of a culture medium based on sterilized calf broth. Pasteur, Thuillier, and Loir went to Bollène in November 1882. The Maucuer couple hosted the Pasteurian delegation and—greatly honored by their presence—did their best to make their stay as pleasant as possible, even gastronomic. Pasteur reported that the quality of the dishes, in particular the truffled guinea fowl, was particularly appreciated. Pasteur quickly wrote to François de Mahy, Minister of Agriculture, to report on the progress of his work, emphasizing the economic impact of the problem. In the Rhône valley, around 20,000 animals had died. In Bollène and in the neighboring villages and castles, the Pasteurians had access to many animals for experimentation. After three weeks on site, Pasteur returned to Paris. The vaccine developed by Pasteur’s team consisted of erysipelas germs attenuated by passing them from rabbits to rabbits. Conversely, Pasteur found that successive passage through guinea pigs or pigeons increased their virulence. The details of the procedure, however, remained vague enough that no one could copy the preparation of the vaccine. Pasteur advised Maucuer: “*The vaccine would be considered of little value if we gave it for free. It will be delivered to you by Mr. Boutroux, 28 rue Vauquelin, at 0 fr 20 centimes per pig. You will charge your work as it fits you*. *However, I believe you would be wrong to charge a high price*” [78]. While some disappointments were reported with some animals that died after vaccination, overall, it was a huge success both in Vaucluse and in various regions of France. In 1892, Loir and Chamberland reported that the death rate was 1.07% for 57,900 vaccinated animals. Pasteur obtained from the Minister of Agriculture that Maucuer would be awarded the Legion of Honor. Since then, an Achille Maucuer Avenue has existed in Bollène, and of course in this same town, a bronze bust of Pasteur was inaugurated in 1924. As in Melun, in 1943 the bust was melted down under the Vichy regime. Rebuilt in 1945, the base and the bust regained their place in the city center of Bollène in 2017.

### 7.4. Rabies

Pasteur’s fourth vaccine was undoubtedly the one that contributed the most to his fame, but it was also a hot topic of contradictory debate. What was the motivation that led Pasteur to work on rabies? Pasteur was a hero among breeders and veterinarians, but tackling a human disease would have much more prestige, more repercussions in the medical world, and in the general population. The choice of rabies may be intriguing because it was an epiphenomenon compared to the mortality resulting from the other infectious diseases that were rife at the time. No doubt he wanted to avoid German competition, which was on many fronts but not that of rabies. What characterizes rabies is the long delay between the bite (assumed to have been given by a rabid animal) and the onset of the disease, at least if the bite was on the extremities of a limb. This delay allowed Pasteur to carry out his inoculations and hope for the establishment of immune protection before the onset of the disease. 

Once again, there are forgotten precursors. Pierre Victor Galtier (1846–1908) was a professor at the veterinarian school of Lyon holding the chair of pathology of infectious diseases (Figure 3). In 1879, he demonstrated the transmissibility of rabies from dogs to rabbits [79]. This key information was then used by Pasteur: rabbits could be a source of rabies virus, and the rabbits used for tests developed the disease very quickly. Roux improved the model by proposing an intracerebral inoculation. On 1 August 1881, Galtier reported to the Academy of Sciences the success of his rabies vaccination [80]: “*I injected rabies saliva into the chinstrap of the sheep seven times, without ever getting rabies; one of my test subjects has since been inoculated with rabid dog slime, and for over four months after this inoculation, the animal has always been well; it seems to have acquired immunity. I inoculated it again two weeks ago by injecting it eight cubic centimeters of rabies saliva into the peritoneum, it is still doing well*”. In total, he injected the rabies virus into the blood stream of nine sheep and one goat. He then injected the deadly virus into these animals and ten control animals. The ten vaccinated animals survived, and the ten control animals perished. In 1886, Galtier published a work entitled: “*Rabies considered in animals and in humans from the point of view of its characteristics and its prophylaxis*”. Even though a bust of Pierre Victor Galtier by Louis Prost can be seen at the veterinary school in Lyon, very few remember his original work.

In addition to Galtier, we should also mention among the precursors Pierre-Henri Duboué (1834–1889), a doctor trained in Paris and member of the Academy of Medicine, who practiced in Pau (Figure 3). On 12 January 1881, he sent his book to Pasteur [81] in which he reported his discovery that the progression of the rabies virus takes place through the peripheral nerve fibers to the central nervous system and not through the blood, at a time when Pasteur was still looking for the viruses in the bloodstream. In 1887, Duboué wrote a new book in which he stated [82]: “*I come with this work, to defend my unjustly unrecognized rights, on the subject of the progresses made in recent years on the great question of rabies, progresses which I can strongly affirm to have been prepared by my own research*”. Decidedly, it was not good to be in the shadow of Pasteur’s work. Later, he added: “*To make it clear the full extent of the denial of justice to me contained in Pasteur’s communication, I must indicate here the reason which gave a whole new direction to the researches of M. Pasteur.” […] “No civet without hare […] Similarly, no preventive treatment possible with attenuated viruses, without the prior culture of the rabies virus, and no culture of the latter, without knowledge of the tissues or organs where this virus resides*” wrote appropriately Duboué.

Pasteur experimented with his rabies vaccine in humans before he had accumulated sufficient evidence of the efficacy and safety of his vaccine. Ethics in Pasteur’s time were obviously not as scrupulous as that of the twenty-first century, as illustrated by his request in 1884 to the Emperor of Brazil to be allowed to test his vaccine on prisoners in exchange for their freedom [83]. In his book and after examining Pasteur’s notebooks, Geison makes a damning observation [3]. Before the first attempts on humans, between August 1884 and May 1885, experiments involved 26 dogs bitten by rabid dogs with three different vaccine approaches. The overall success rate was 62%. However, none of them correspond to the one used on Joseph Meister. This is undoubtedly one of the reasons why his most loyal collaborator, Dr Roux, refused to test the vaccine in humans on which he himself had been working, and it was Joseph Grancher who performed the injections. As a source of attenuated viruses, it was Roux’s idea to dry out the spinal cords of rabbits that had succumbed to rabies, hung in vials, while Pasteur had the idea to add potash to accelerate the drying. Pasteur’s laboratory notebooks reveal that Joseph Meister was not the first human to be treated with Pasteur’s rabies vaccine. The very first rabies vaccination was carried out on 2 May 1885 on a patient of Dr. Georges Dujardin-Beaumetz (1833–1895), member of the Academy of Medicine, at Necker hospital, named Mr. Girard (61 years old), who had been bitten on the knee by a rabid animal. The treatment was initiated with two injections twelve hours apart. However, the treatment was stopped by the hospital authorities who had consulted the Ministry of Public Health. On May 3rd, Girard’s conditions deteriorated with tremors that lasted three days. On May 7th, the patient was much better, and a fortnight later, the patient was discharged from the hospital. Doubt persisted as to the nature of this first patient’s illness. The second injection took place on 22 June 1885 at the Saint-Denis hospital, where an eleven-year-old girl, Julie-Antoinette Poughon, was vaccinated. Unfortunately, she died the next day, suggesting that the administration of the vaccine was too late. The best-known inoculation took place on 6 July 1885 on young Joseph Meister (1876–1940), aged 9, who came with his mother from Alsace. Administration of the vaccine, as defined by Pasteur, consisted of a succession of inoculations with the desiccated spinal cords of rabid rabbits by injecting increasingly virulent virus preparations until the fully active virus was injected. The experimental approach with parched spinal cord was initially started on 28 May and 3 June in 20 dogs and repeated on 25 and 27 June in 20 new dogs. This is to say whether on July 6th Pasteur had little data to ensure the efficacy and safety of his protocol. Pasteur notes in his notebook: production of the refractory state on a child very dangerously bitten by a rabid dog. Pasteur is aware of the dangerousness of his treatment: “*Joseph Meister therefore escaped not only the rage caused by the bites, but also the one I injected into him to control immunity*” [84].

The second vaccination was carried out on 20 October 1885 in a young 15-year-old shepherd, Jean-Baptiste Jupille (1869–1923). Meister, like Jupille, was infinitely grateful and became a guardian of the Institut Pasteur. Meister committed suicide when the Nazi army entered Paris. Another thing the two young people had in common was that there was a lack of evidence that they were actually bitten by rabid dogs. The dog that bit Meister was killed and autopsied; finding wood debris in his stomach was the only evidence that he would have had rabies. Yet, Michel Peter (1824–1893), member of the Academy of Medicine, reminded his colleagues: “*In the past, you remember, any dog in whose stomach one found foreign bodies: wood, straw, etc., was famous enraged; this proof is abandoned*” [85]. As for Jupille, it was not the dog who attacked the child but the child who attacked the dog by rushing towards him with his whip (which Pasteur knew). The animal defended itself and bit young Jupille’s hand. The latter tied the dog’s mouth with the rope of his whip and threw it into the river. The statue which stands in the grounds of the Institut Pasteur, where one sees the young Jupille fighting with a dog to protect his little comrades from the attack of a mad dog, helped to propagate the legend.

During the course of Jupille’s vaccination, Grancher pricked himself in the thigh with the needle of a syringe filled up with four-day-old spinal cord, that is to say containing virulent virus. A full vaccination process was then necessary for Grancher. Pasteur asked to be inoculated as well. Grancher refused, as did Loir, and in doing so disobeyed Pasteur for the first time. Then, Adrien Loir and Eugène Viala (1858–1926), a laboratory technician, got vaccinated as well [68].

On 26 October 1885, Pasteur presented his successes to the Academy of Sciences and its president, Henri Bouley (1814–1885), which was hailed a memorable moment in the history of medicine and forever glorious in French science. The next day, Pasteur presented the same results to the Academy of Medicine, hailed once again as a most memorable moment in the history of the conquests of science and in the annals of the Academy. The international impact was arguably as high as Pasteur had hoped. As early as the fall/winter of 1885, people came from far away to receive the life-saving treatment: from Russia, the nineteen muzhiks of Smolensk (fifteen of whom were rescued), who had been welcomed at the railway station by the Baron Arthur Pavlovitch de Mohrenheim, ambassador of Russia in Paris, or from America, the four boys from New Jersey. Robert M. McLane (1815–1898), ambassador of USA, offered a banquet to glorify Pasteur. These successes played an essential role in the creation of the Institut Pasteur, inaugurated on 14 November 1888 [86].

Léon Perdrix (1859–1917), a former student at ENS and associate preparer in Pasteur’s laboratory, published the results of the first years of vaccination [87]: from 1886 to 1889, 7893 people (including 15.9% foreigners) were treated, and the mortality was only 0.67%. Admittedly, the mortality from rabies was greater than 98%, but how many of the people treated had actually had rabies? Decourt says he has verified and estimated the number of cases of rabies in France during the years 1850 to 1876 at 28.5 cases per year. It emerged that a number of people were treated even though they had not been bitten by rabid dogs [27].

Despite the aura of Louis Pasteur’s vaccination against rabies, a few clouds gathered in the sky of the glorious hero. There is first the “Jules Rouyer affair”, when a ten-year-old boy was bitten on the arm by an unknown dog through his overcoat on 8 October 1886. Pasteur was on vacation in Bordighera on the Italian Riviera, and it was Andrien Loir who took over the vaccination. Rabies inoculations began on 20 October, carried out daily for twelve days. Sadly, the child died on 26 November. Due to the father, Édouard Rouyer, having lodged a complaint, an autopsy was performed in Loir’s presence by Brouardel and Grancher, who took the child’s medulla oblongata and sent it to Roux so that he could inoculate two rabbits. The result was not long to come, and both rabbits quickly died of paralytic rabies. Roux and Brouardel perjured themselves in court, claiming that the rabbit tests had been negative and that the child had not died of rabies but of a uremic attack [7]. By doing so, they felt they were acting for the benefit of mankind by saving vaccination. The risk/benefit ratio of such a revelation was recognized by Roux himself. However, not all were convinced, in particular Michel Peter, who considered that the child had indeed died of rabies. He regularly opposed Pasteur, especially since he also witnessed another case of death despite (or because?) of the vaccine, that of a young man of twenty, called Réveillac, who died of rabies after receiving treatment. Peter declared: “*To amplify the benefits of his method and to mask its failures, Mr. Pasteur has an interest in making the annual mortality rate from rabies in France believed to be higher. But these are not the interests of the truth. Do we want to know, for example, how many individuals in 25 years have died of rabies in Dunkerque? He died of it: one… And do we want to know how many died in this city in a year, since the application of the Pasteurian method? He died: one*”. However, Pasteur considered Peter’s words null and void. Among the failures, let us also note the case of Hayes St Leger, fourth Viscount Doneraile (1818–1887). In Ireland at the time of the last outbreak of rabies, Lord Doneraile and his coachman Robert Barrer were both bitten by a rabid fox on 13 January 1887. Lord Doneraile suffered severe, multiple, and deep bites on both hands. They went to Paris to receive the full treatments between 24 January and 21 February. Unfortunately, Lord Doneraile finally died of rabies on 26 August 1887 as a result of fox rabies or the inoculation during the treatment. Pasteur dealt with another opponent, Anton von Frisch (1849–1917), an Austrian urologist who had nevertheless come to train with him. In 1887, he published a work entitled “*The treatment of rabies disease: an experimental critique of Pasteur’s method*”, in which he questioned the reliability and relevance of Pasteur’s vaccine approach.

The discovery that rabies was due to a virus was made in 1903 by Paul Remlinger (1871–1964), director of the Imperial Bacteriology Institute of Constantinople [88]. At the beginning of the 20th century, in Italy, Claudio Fermi (1862–1952), a doctor who worked at the Institute of Hygiene in Rome, questioned the preparation of the vaccine. He applied the Toussaint’s method, exposure to phenol, developing a vaccine that was simpler, more effective, and above all safer, without risk of transmission since the virulence was essentially eliminated. The poor value of the vaccine as it had been defined by Pasteur from dehydrated spinal cord was demonstrated by one of his heirs within the institution he had created: Pierre Lépine (1901–1989), a physician who had joined Professor Constantin Levaditi (1874–1953) in 1927. Lépine was director of the Institut Pasteur in Athens from 1930 to 1935, then head of the virology department at the Institut Pasteur in Paris from 1940 to 1971. In 1937, Lépine undertook a comparative study of the rabies vaccines of Pasteur and Fermi. He demonstrated by injecting 40 rabbits that the protective power of the Pasteur vaccine was 35%, while tested on 52 rabbits, Fermi’s vaccine reached a protection rate of 77.7% [89].

## 8. Concluding Remarks

After Jenner, Béchamp, Toussaint, and Galtier, Pasteur allowed vaccination to acquire its credentials. However, it was not until the experiments of Emil von Behring (1854–1917), Shibasaburo Kitasato (1853–1931), and Paul Ehrlich (1854–1915) that science could fully understand the exact nature of the immune host response [90,91]. Pasteur, as a microbiologist, conceived of the protection acquired by vaccination by attenuated bacteria as a consumption of the nutritional requirements needed for the growth and survival of the microbe, just as a culture media contained only trace amounts of vital nutrients. Thus, the host would not support the growth of a subsequent infection by the same microbe [92]. Pasteur admitted that the use of dead germs for vaccines did not fit with his own explanation.

In his obituary published in *Science* in 1895, the American bacteriologist H.W. Conn, director of the Cold Spring Biological laboratory, wrote: “*Pasteur is regarded as the father of modern bacteriology, but we must remember that he was not a pioneer in these lines of work. There was hardly a problem that he studied which had not been already recognized, and even studied to a greater or less extent by his predecessors*”, but nicely adding “*Others discovered facts, Pasteur determined laws*” [93]. Fifty years later, when a special exhibition devoted to Louis Pasteur was organized in London (1947), Alexander Fleming paid tribute to the great scientist. He cited many of those who, with Pasteur, contributed to the fight against microbes, but he failed to mention Béchamp, Toussaint, Feltz, Duboué, or Galtier, illustrating that Pasteur’s efforts to minimize the role played by his precursors had been successful. The legend was written and even a leading figure would not dare to flout it [94].

## Figures and Tables

**Figure 1 biomolecules-12-00596-f001:**
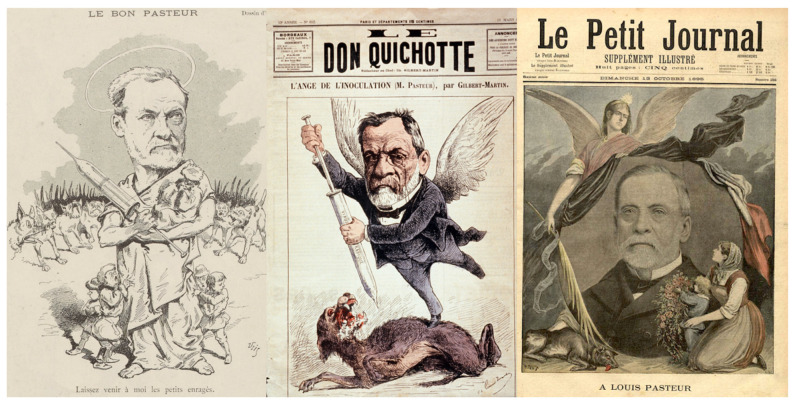
Left: Pasteur in the French Press seen as a lay saint (*Le Courrier Français*, 4 April 1886); center: as an angel fighting rabies (*Le Don Quichotte*, 13 March 1886); right: as a revered icon after his death (*Le Petit Journal*, 13 October 1895). (© Institut Pasteur, Musée Pasteur).

**Figure 2 biomolecules-12-00596-f002:**
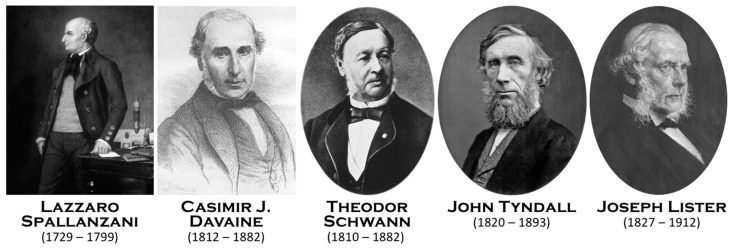
The main predecessors recognized by Louis Pasteur (Spallanzani, Davaine, and Schwann) and his main supporters (Tyndall and Lister) (© Institut Pasteur, Musée Pasteur; © Wikipedia; © Collection of Pauls Stradiņš, Museum of History of Medicine, Riga, Latvia).

**Figure 3 biomolecules-12-00596-f003:**
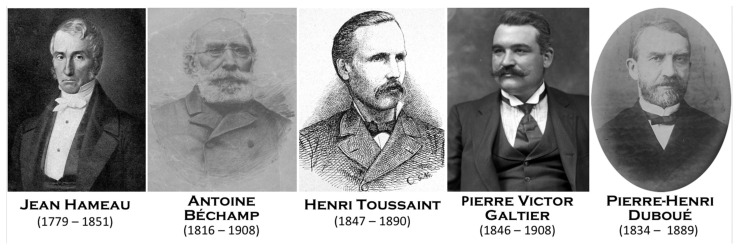
Louis Pasteur faced numerous precursors, opponents, and competitors. Some were wrong, but a few, particularly Hameau, Béchamp, Toussaint, Galtier, and Duboué, were right despite being unknown or poorly recognized by Louis Pasteur (© Wikipedia/© https://gw.geneanet.org accessed on 19 March 2022).

**Figure 4 biomolecules-12-00596-f004:**
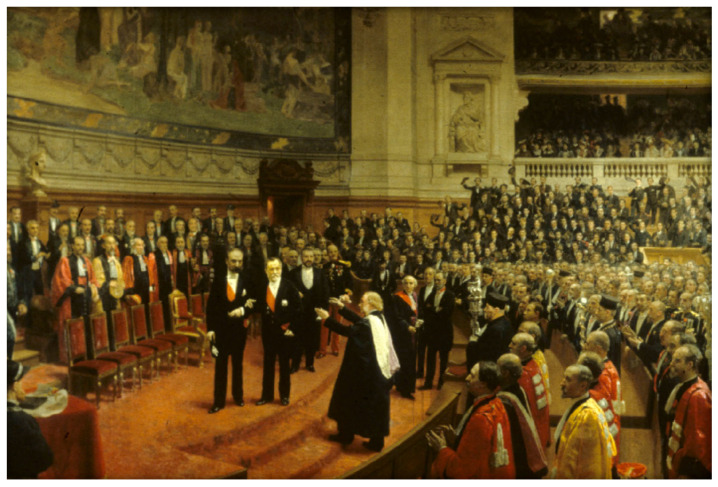
On 27 December 1892, for his 70th birthday, the international scientific community celebrated Pasteur’s “jubilé”. The reception took place in the great amphitheater of “la Sorbonne”. On the picture, one sees the president of France, Sadi Carnot, helping Pasteur to walk and Lister climbing a few steps to congratulate Pasteur. Oil on canvas by Jean-André Rixens (1902). © Institut Pasteur/Musée Pasteur.

**Table 1 biomolecules-12-00596-t001:** Main contributions of Louis Pasteur.

1848–1858	Studies on molecular chirality: crystallography of tartaric and paratartaric acid
1857–1879	Studies on fermentation; First patent on alcoholic fermentation (1857)
1861	Discovery of anaerobic bacteria
1861–1879	Refutation of the theory of spontaneous generations. Discovery of germs
1863–1873	Studies on diseases of wine, vinegar, and beer
1865	Pasteurization of wine; Patent on wine preservation
1865–1870	Study on the diseases of silkworms
1871	Patent on beer preparation and preservation
1877	First observation of antibiosis
1877–1881	Studies on infectious diseases (anthrax, puerperal sepsis, boils)
1878	Demonstration in a vineyard that isolation of grapes from environmental air prevents fermentation in the further wine-making process
1880	Co-discovery with Alexander Ogston (UK) of *Staphylococcus aureus*
1881	Co-discovery with George M. Sternberg (USA) of *Streptococcus pneumoniae*
1880–1885	Preparation of vaccines (fowl cholera, anthrax, pig erysipelas, rabies)
1887	First bacteriological war: elimination of rabbits by *Pasteurella multocida* over the cellar of Champagne of Mrs. Pommery (Reims)

**Table 2 biomolecules-12-00596-t002:** Main steps of Louis Pasteur’s life and career.

27 December 1822	Birth in Dôle (Jura) (third child of Jean-Joseph Pasteur (1791–1865) and Jeanne-Étiennette Roqui (1793–1848)
1827	The family moved to Arbois
1831–1843	Studied in Arbois, Besançon, Dijon, and Paris
1844–1847	Studied at Ecole Normale Supérieure (ENS, Paris)
1846	“Agrégé préparateur” at ENS
1847	Thesis for his Doctorat ès-Sciences (physics and chemistry)
1848–1853	Taught physics in high school in Dijon and chemistry at the University of Strasbourg
29 May 1849	Married Marie Laurent, daughter of the Strasbourg university’s rector
1850	Birth of Jeanne, first child (deceased in 1859, 9 ½ years)
1851	Birth of Jean-Baptiste, second child (deceased in 1908)
1853	Birth of Cécile, third child (deceased in 1866, 12 ½ years)
	Knight of the Légion d’Honneur
1854	Professor of chemistry and dean of the faculty of sciences of Lille
1857	Failure of his application to the Academy of Sciences
1857–1867	Administrator and director of scientific studies at ENS
1858	Birth of Marie-Louise, fourth child (deceased in 1934)
	Set up his research laboratory in the attics of ENS
1862	Election at the French Academy of Sciences
1863	Birth of Camille, fifth child (deceased in 1865, 2 years)
	Professor of geology, physics, and applied chemistry at the School of Fine Arts
1867–1888	Director of a laboratory at ENS
1867–1872	Professor, chair of organic chemistry at the Sorbonne
1868	First severe brain stroke that paralyzed his left side
1873	Election at the French Academy of Medicine
1875	Failure to be elected Senator for Jura
1879	His daughter Marie-Louise married René Valéry-Radot (1853–1933)
1881	Election at the French Academy; Great Cross of the Légion d’honneur
1888–1895	Director of Institut Pasteur
28 September 1895	Death in Institut Pasteur annex (Marnes la Coquette)
26 December 1896	The coffin of Louis Pasteur was transferred in the crypt of Institut Pasteur

**Table 3 biomolecules-12-00596-t003:** Some of the precursors who, before Louis Pasteur, proposed the germ theory and/or refuted the concept of spontaneous generation.

Before JC	Marcus Terentius Varro (Varron) (116 BC–27 BC) (Roman)
1st century	Galen of Pergamon (129–216) (Greece)
1546	Girolamo Fracastoro (1483–1553) (Italy)
1658	Athanasius Kircher (1601 or 1602–1680) (Germany)
1663	Robert Boyle (1627–1691) (Ireland)
1668	Francesco Redi (1626–1697) (Italy)
1714	Nicolas Andry de Bois-Regard (1658–1742) (France)
1718	Louis Joblot (1645–1723) (France)
1720	Benjamin Marten (1690–1752) (UK)
1721	Jean-Baptiste Goiffon (1658–1730) (France)
1762	Marcus Antonius von Plenčič (1705–1786) (Austria)
1765	Lazzaro Spallanzani (1729–1799) (Italy)
1836	Theodor Schwann (1810–1882) (Germany)
1836	Franz Schulze (1815–1921) (Germany)
1837	Jean Hameau (1779–1851) (France)
1839	Sir Henry Holland (1788–1873) (UK)
1840	Jakob Henle (1809–1885) (Germany)
1844	Agostino Bassi (1773–1856) (Italy)
1846	Gideon Algernon Mantell (1790–1852) (UK)
1866	Auguste Chauveau (1827–1917) (France)

## Data Availability

Not applicable.

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
