# Peer review of "Louis Pasteur: Between Myth and Reality"

_biomolecules, 2022, doi:10.3390/biom12040596_

Round 1

Reviewer 1 Report

The review article entitled “Louis Pasteur: Between myth and reality” by Jean-Marc Cavaillon and Sandra Legout is a well compiled brilliantly well-written biography portraying the most realistic aspects of the academic and professional life of one of the most iconic figures in the field of microbiology in the world’s history. I find the article very well balanced since it makes an objective critique of the person without fully discrediting the true ingenuity and talents Luis Pasteur possessed. Moreover, this piece timely touches the core of scientific ethics and plagiarism, as well as provides pragmatic example as to how these could be easily corrupted and wrongfully fostered by having the right economic and political alliances. These undeniable facts make this article a highly relevant piece. Vices like unethical behavior and plagiarism remain in most fields of contemporary science and culture. Thus, it is critically important to raise awareness about them as a countermeasure for a lack of adequate enforcement and accountability. This article does that!!..

Minor comments.

I found a little hard to distinguish when authors were referring to information found in Pausteur’s notes, that when they were just telling anecdotical information (excluding actual quotations that are perfectly defined throughout the text), since there are large written segments throughout the text that have no citations.

Reviewer 2 Report

This is a very interesting and challenging article that describes the scientific facts and myths related to each of the discoveries attributed to Louis Pasteur. The historical work is precise, well-documented and controversies are clearly stated. The article is well written, didactic and fascinating, even if it makes Pasteur fall from his pedestal… Overall, this review is excellent.

Minor comments:
Table 2: 1863, Birth of Camille fifth (and not second) child…
Paragraph 5.3, line 7: On January 7th, 1884, “He” should be written “he”sent.
Paragraph 8, line 2 : But (one… to be deleted)) it was…